

# Investigation of the long-term variations in hydro-climatology of the Dinder and Rahad basins and its implications on ecosystems of the Dinder National Park, Sudan

Khalid Hassaballah[1, 2, 3,*], Yasir Mohamed[1, 2, 3], Stefan Uhlenbrook[1, 2]

[1]UNESCO-IHE Institute for Water Education, P.O. Box 3015, 2601DA Delft, The Netherlands

[2]Delft University of Technology, Faculty of Civil Engineering and Applied Geosciences, Water Resources Section, Stevinweg 1, P.O. Box 5048, 2600 GA Delft, The Netherlands

[3]Hydraulics Research Centre, P.O. Box 318, Wad Medani, Sudan

*Correspondence to*: Khalid Hassaballah (k.hassaballah@yahoo.com)

**Abstract.** Hydro-climatic variability plays a pivotal role in structuring the biophysical environment of riverine and floodplain ecosystems. Variability is natural, but can also be enhanced by anthropogenic interventions. Alterations of hydro-climatic variables can have significant impacts on the ecohydrological functions of rivers and related ecosystems. Loss of biodiversity and degradation of ecosystems have caused increasing concern about the current situation of the Dinder and Rahad River basins (D&R), particularly the ecosystems of the Dinder National Park (DNP). However the causes are not yet fully understood. Conservation of the DNP ecosystems for direct and indirect human benefit is one of major challenges facing the country.

This paper examines the long-term variations of streamflow, rainfall and temperature over the D&R and its implications on DNP ecosystems. Statistical tests of Mann-Kendall (MK) and Pettitt were used. The analysis was carried out for twelve precipitation, one temperature, and two streamflow gauging stations over different time periods. Streamflow characteristics of magnitude, duration, timing, frequency and rate of change in flow that likely impact the ecological functions of the ecosystem of the DNP, were analysed using the Indicators of Hydrologic Alterations (IHA).

The MK test showed statistically significant increasing trends of temperature. The mean annual and monthly mean precipitation showed no significant change. Streamflow of the Rahad River showed a significant increasing trend in annual and monthly means at Al-Hawata station, while no significant trend in Dinder River flows at Al-Gewisi station could be observed. However, the Dinder river showed significant decreasing trend in maximum annual and monthly mean and maximum flow during August (month of high flow), and increasing trend during November (month of low flow). The IHA analysis indicated that the Rahad River flow was coupled with significant upward alterations in some of the hydrological indicators. In contrast, the Dinder River flow was coupled with significant downward alterations. This alterations in Dinder river flow are likely affect the ecosystems in DNP negatively. Alterations in magnitude and duration of the annual flood peaks





that reduce the amount of water flowing to the river-floodplain, may diminish the production of native flora and fauna, and animal migration that may be linked to floodplain inundation.

**Keywords**: Dinder and Rahad catchment, time series analysis, IHA, Hydro-climatic variables

# 1    Introduction

Hydro-climatic variability plays a pivotal role in structuring the biophysical environment of riverine and floodplain ecosystems. Variability is natural, but can also be enhanced by anthropogenic interventions. Alterations of hydro-climatic variables can have significant impacts on the ecohydrological functions of rivers and related ecosystems.

The headwater catchments of the Dinder and Rahad Basins (D&R) generate over 7% of the Blue Nile basin's annual flow. The Rahad River supplies water to the Rahad Irrigation Scheme, while the Dinder River is the main source of water for the diverse ecosystem of the Dinder National Park (10,291 km$^2$). However, during recent years, the Dinder River has experienced significant changes of floodplain hydrology and water supply to local wetlands (mayas). The causes are not fully known. This has significant implications on the ecosystem functions and hence the services of the Dinder National Park (DNP). Therefore,

understanding of climate variability/change and its hydrological impacts is essential for water resources development, as well as for sustainable ecosystem conservation in the DNP.

Trend analysis for hydrological and meteorological time series is an important and common method for understanding climate variation and its impacts on water resources (Burn and Hag Elnur, 2002; Kahya and Kalaycı, 2004). The existence of a trend in hydrologic time series can be explained by the change in streamflow (e.g. Lins and Slack, 1999; Woo and Thorne, 2003;

Cigizoglu et al., 2005). It can also be explained by changes in precipitation (e.g. Lettenmaier et al., 1994; Rodriguez-Puebla et al., 1998; Partal and Kahya, 2006). Temperature trends were analysed to understand links to hydrology as a proxy for changes of evapotranspiration (e.g. Ghil and Vautard, 1991; Stafford et al., 2000; Vinnikov and Grody, 2003).

A good number of studies used the Mann-Kendall (MK) test (Mann, 1945; Kendall, 1975), to identify hydro-climatologic trends. Tesemma et al. (2010), analysing the trends of rainfall and streamflow over a 40 year period (1963-2003), showed no

change of precipitation over the Blue Nile basin. Streamflow analysis for Bahir Dar and Kessie at the upper portion of the Blue Nile basin, and El Diem at the border between Sudan and Ethiopia showed that the annual streamflow did however indicate increased flow in the upper Blue Nile, but not at El Diem. Using MK and Pettitt tests, Gebremichael et al. (2013), found no significant change of the annual precipitation over the Upper Blue basin between the 1970s and the beginning of the 21[st] century. Nevertheless, both tests showed a statistically significant increasing trend of streamflow during the long rainy season

(June-September) and the short rainy season (March-May), and a decreasing trend in the dry season (October-February) streamflow. The annual streamflow has increased significantly during the period (1971-2009). Since the Upper Blue Nile basin is neighbouring the D&R, similarities of catchment characteristics could be expected. Tekleab et al. (2013), studied the trends of rainfall, temperature and streamflow within the Abbay/Upper Blue Nile basin. The results showed statistically significant





increasing and decreasing trends in the streamflow. Temperature showed increasing trends in most of the studied stations. In contrast, rainfall did not show any significant trends.

Recently, Masih et al. (2014), who reviewed droughts on the African continent, stated that the available evidence from the past clearly shows that the continent is likely to face extreme and widespread droughts in future. They speculate that drought challenge is likely to aggravate because of slow progress in drought risk management, increasing population and demand for water, and degradation of land and environment. In contrast, Bashir et al. (2016), assessing the impacts of future climate change (2020s, 2050s, and 2080s) on the Dinder River flow and its possible implications on the DNP ecosystems, found that the climate will become warmer and wetter.

Thus, it has been shown that a variety of probable climatic impacts on hydrologic system of the D&R is likely to happen. Therefore, the objective of this study is to investigate the long-term variations of streamflow, rainfall and temperature over the D&R and its implications on the DNP wetlands ecosystems. Streamflow regime is essential in sustaining ecological integrity of river systems (Poff et al., 1997). Therefore, The IHA approach was then applied to support the MK test and to analyse the essential characteristics of the streamflow likely to impact ecological functions in the D&R basin, including: flow magnitude, flow timing, flow frequency, flow duration and rate of change in river flows. Understanding the level to which the streamflow has changed from its natural conditions is crucial for developing effective management plan for ecosystem restoration and conservation.

LULC changes has widely reported in the literature (e.g. Zeleke and Hurni, 2001; Tesemma et al., 2010; Teferi et al., 2010; Gumindoga et al., 2014) especially deforestation and expansion in crop area after the severe drought in East Africa during mid 1980's. Therefore, we assumed the LULC changes occurred around early 1990's in the upper Dinder and Rahad. We have made the subdivision from 1992 to obtain equal number of years before and after the changes for the IHA statistical comparison.

The hydrological effect of LULC changes on river flow within the neighbouring catchment especially during mid 1980s has reported in the literature (e.g. Gumindoga et al., 2014). To evaluate the current Dinder and Rahad Rivers hydrology relative to historical conditions, the natural ranges of flows variations for both Rivers have been characterized using the IHA for comparing two periods (1972-1991) and (1992-2011), hereafter defined as pre and post-impact, respectively. Natural temporal variability of flow data were analysed from Al-Gewisi station on Dinder River and Al-Hawata station on Rahad River.

## 2 Study area

The D&R are the lower sub-basins of the Blue Nile River basin (BN). The BN collects flows of eight major tributaries in Ethiopia beside the two main tributaries in Sudan: the Rahad and the Dinder Rivers. Both tributaries are fast flowing streams during the flood season and derive their water from the runoff of the Ethiopian highlands about 30 km west of Lake Tana (Hurst et al., 1959). The Dinder River joints the Blue Nile at the village Al-Rabwa, 64 km downstream of the Sennar reservoir, while the Rahad River joins the Blue Nile at the village Abu Haraz below Wad Medani town (Fig. 1). The elevation ranges



from 2731 m at the Ethiopian plateau to 389 m at the outlet of the Rahad River. The basin boundary and the streams system have been delineated from a 90 m x 90 m digital elevation model database of the NASA Shuttle Radar Topographic Mission (SRTM) acquired from the Consortium for Spatial Information (CGIAR_CSI) website (http://srtm.csi.cgiar.org).

The hydrology of the D&R is complex, with varying climate, topography, soil, vegetation and geology as well as human interventions. The D&R basin total area is some 77 486 km$^2$, composed of Dinder sub-basin (34 964 km$^2$) and Rahad sub-basin (42 540 km$^2$). The mean annual flow is about 2.797 x 10$^9$ and 1.102 x 10$^9$ m$^3$/y for Dinder and Rahad, respectively with maximum flow during August/September. Monthly precipitation records indicate a summer rainy season, from June to September. The rains during this season account for nearly 90% of total annual precipitation in the lower part of the basin, while in the Ethiopian highlands, approximately 75% of the annual precipitation falls during these months (Shahin, 1985). The

annual rainfall reaches 1400 mm/y in the Ethiopian highlands near Lake Tana, but only 440 mm/y at Sennar station at the lower part of the basin. The mean annual temperature over the Ethiopian plateau does not exceed 20 °C, while it exceeds 30 °C at the outlet in Sudan. The mean annual evaporation follows spatial pattern of temperature. It reaches 1150 mm/y over the highland, but exceeds 2500 mm/y in the lower part of the basin in Sudan (Block, 2007).

The main soil types in the D&R according to the FAO classification are: vertisols 71%, luvisols 9%, nitisols 8%, leptosols

5%, cambisols 4%, alisols 2% and fluvisols 1%. The vegetation cover is characterized by grasslands, shrublands, croplands, and woodlands.

## 2.1   Dinder National Park (DNP)

The DNP is located on both sides of Dinder river in the South-east of Sudan close to the Sudanese-Ethiopian boundary between latitude 11°00' and 13°00' N and longitude 34°30' and 36°00' E. It was recognised as a national park in 1935 after the 1933

London Convention for the conservation of African flora and fauna (Dasmann, 1972), and declared a biosphere reserve in 1979 (AbdelHameed, 1998). The DNP is a vital ecological area in the arid and semi-arid Sudano-Saharan region. The water system of the park consist of both the Dinder River in the middle of the park and the Rahad River on the Northern border of the park and their tributaries and mayas. "Mayas", a local name for floodplain wetlands that are found on both sides along the Dinder River. According to DNP authority, there are more than 40 mayas within the Dinder and Rahad floodplains. Mayas are

the main source of food and water for wildlife (herbivores) in the park, especially during the dry season which extend from November to June (AbdelHameed et al., 1997). The park has an economic, environmental and social benefits, and provides a huge range of ecosystem services to the communities living within and around the park. The provided services can be grouped as provisioning (e.g., food (fish and honey), fuel wood and medicines that extracted from biota, regulating (flood, climate and groundwater recharge), high potential opportunity for tourism and educational (e.g. attractive place for local people and

foreigner and opportunity for research and training), and supporting services (e.g. shelter for wildlife).



## 2.2    Mayas ecosystem

Dasmann (1979), categorised the DNP vegetation into four main classes: wooded grassland; open grassland, woodland and riverine forest. While, the vegetation assessments by Hakim et al. (1978), and Abdel Hameed et al. (1996a), documented three categories of ecosystems, namely the *Acacia seyal-Balanites aegyptiaca* ecosystem, the riverine ecosystem and the mayas

ecosystem. The mayas support large communities of inhabitants of wildlife animals throughout the dry season and a smaller number during the wet season. Yousif and Mohamed (2012), reported that waterbuck (*Kobus defassa harnieri*), reedbuck (*Redunca bohor cottoni*), tiang (*Damaliscus korrigum tiang*), buffalo (*Syncerus caffer aequinoctialis*), oribi (*Ourebia ourebia montana*), roan antelope (*Hippotragus equinus bakeri*), warthog (*Phacochoerus aethiopicus aelinani*), and bushbuck (*Tragelaphus scriptus bor*), are the major herbivores that inhabit the DNP, while other animals such as baboon (*Papio anubis*)

and red hussar monkey (*Erythrocebus patas*) are numerous. They have also stated that the major predators are lion (*Panthera leo leo*), striped hyena (*Hyaena hyaena dubbah*) and spotted hyena (*Crocuta crocuta fortis*).

Mayas are habitats for various birds. Since the DNP lies on the way of migration of African birds (Abdel Hameed and Abdelhafes 2003), mayas are also provide a shelter for around 250 species of migratory birds.  The mayas habitats are diverse and are a main breeding place for fishes, water dwelling insects, amphibian and micro fauna which significantly improve the

biodiversity of these mayas wetlands (Abdel Hameed and Abdelhafes 2003). Mayas support protection of fishes during the dry season and thus are a valuable reserve when the next flood comes and connects the mayas to the main channel of Dinder and then to the Blue Nile and River Nile. Many types of fishes of the River Nile are represented in mayas. According to HCENR-WRC (2002), out of the 115 species of fish counted in the Nile, 32 fish species are found in the mayas and the remaining pools in the Dinder River during the dry season (November-June). Each maya is a habitat for species that varies in

in both quantity and quality.

Loss of biodiversity and degradation of mayas ecosystems have caused increasing concern about the current situation of the DNP, however the causes are not yet fully understood. Conservation of the DNP ecosystems for direct and indirect human benefits is one of major challenges facing the country.

## 3    Methods and data used

## 3.1    Trend detection tests

In this study, the non-parametric Mann-Kendall (MK) and Pettitt tests (Mann-Kendal, 1975 ; Pettit, 1979) were applied to analyse the trends and the changing points of three hydro-climatic data time series of streamflow, precipitation and temperature. Trends have been assessed in different time periods and varying lengths based on data availability.

The MK statistic is given by:





$$S = \sum_{i=1}^{n-1} \sum_{j=i+1}^{n} \mathrm{Sgn}(X_j - X_i) \qquad (1)$$

Where $S$ is the MK statistic, $X_i$ and $X_j$ are the observations with $j > i$, $n$ is the time series data set length, and the sign function is given by:

$$Sgn(\theta) = \begin{bmatrix} +1 & \text{if} & \theta > 0 \\ 0 & \text{if} & \theta = 0 \\ -1 & \text{if} & \theta < 0 \end{bmatrix} \qquad (2)$$

The variance Var(S) and the standard normal variate Z are calculated with Eqs. (3) and (4), respectively. The trend results in this study have been assessed at 5% significant level.

$$Var(s) = \frac{1}{18}\left[n(n-1)(2n+5) - \sum_{t} t_i(t_i - 1)(2t_i + 5)\right] \qquad (3)$$

Where $t_i$ is the extent of any given tie, and $\Sigma_t$ denotes the summation over all ties. $H_0$ should be accepted if $|z| \leq z\alpha/2$ at the $\alpha$ level of significance.

$$Z = \begin{cases} \dfrac{s-1}{\sqrt{Var(s)}} & \text{if} & s > 0 \\ 0 & \text{if} & s = 0 \\ \dfrac{s+1}{\sqrt{Var(s)}} & \text{if} & s < 0 \end{cases} \qquad (4)$$

The magnitude of the slope β, determined by (Hirsch et al. 1982) is given by:

$$\beta = \mathrm{Median}\left[\frac{(X_{j-} X_i)}{(j-i)}\right] \qquad \text{where } 1 < i < j < n \qquad (5)$$

Where $X_i$ and $X_j$ are the data values at time $i$ and $j$, respectively and $n$ is the length of the whole data set.

The existence of increasing or decreasing trends was tested using the MK test. Then, the Pettitt test was applied to detect the changing points. The Pettitt test is a non-parametric test used to identify a single change-point in the data series if any (Pettitt, 1979). The significance of trends in the dataset is defined as "no significant trend", "significantly increasing or decreasing





trend" based on the defined confidence level of 5%. The MK computes Kendall's statistics (S), Kendall's tau ($\tau$) and MK's Z statistic. Positive Z values indicate an increasing trends whereas negative values indicate a decreasing trends. Finally, a probability (p-value) was computed and compared with the user defined significance level in order to identify the trend of variables.

5   ## 3.2    Indicators of hydrologic alterations (IHA)

The IHA technique is part of the Range of Variability Approach (RVA) developed by Richter et al. (1997). It is used to assess river ecosystem management goals defined based on a statistical representation of ecologically related hydrologic parameters (Richter et al., 1996). These parameters describe five essential characteristics of river flow that have ecological implication (Richter et al., 1996; Poff et al., 1997; Scott et al., 1997). The IHA technique computes 33 hydrologic parameters for each

10   year.

For analysing the alteration between two periods, the RVA described in Richter et al. (1997) was applied using the IHA software developed by The Nature of Conservancy (2009). In RVA analysis, the pre-impact data for each parameter is divided into three categories. In this study, boundaries between categories were defined based on the default percentile values for non-parametric RVA analysis by adjusting the category boundaries 17 percentiles from the median. This ensures that in most

conditions an equivalent number of values will fall into each category and gives three categories of equal size as given in Eq. (6):

$$C_l \leq P^{33} < C_m \leq P^{67} < C_h \qquad (6)$$

Where, $C_l$, $C_m$ and $C_h$ are the low, middle and high categories, respectively. $P^{33}$ and $P^{67}$ are the 33th and 67th percentiles, respectively.

The Hydrologic Alteration (HA) factor is calculated for each of the three categories as given in Eq. (7):

$$HA = \frac{f_o - f_e}{f_e} \qquad (7)$$

Where; $f_o$ is the observed frequency, and $f_e$ is the expected frequency.

Hydrologic Alteration with a positive deviation indicates an increasing in frequency of the value within the target category compared to the pre-impact period, while a negative deviation indicates a decreasing (The Nature Conservancy, 2009).

For assessing hydrologic alteration in the Dinder and Rahad Rivers, the natural ranges of flows variations for both rivers have been characterized using the IHA based on variations in streamflow characteristics between two periods (1972-1991) and



(1992-2011), hereafter defined as pre and post-impact periods, respectively. Natural temporal variability of flow data were analysed from Al-Gewisi station on the Dinder River and Al-Hawata station on the Rahad River.

## 3.3    Hydro-climatic data

The hydro-climatic variables streamflow, precipitation and temperature are selected because of a) the spatially assimilated hydrologic response that they provide, and b) they are the only variables having available long records of data. Temperature was used as a proxy for evapotranspiration. Table 1 shows the available data and their minimum, maximum and mean annual total values.

There are twelve precipitation stations spatially distributed over the study area. Data are monthly. Six stations are in Rahad basin: Gedarif, Gadambaleya, Samsam, Um Seinat, Doka and Al-Hawata. Since there is no station with long records in the Dinder basin, data from four nearby stations were used: Ad Damazin, Abu Naama, Um Benien and Sennar. The same is true for the upper part of the catchment, so data from two nearby stations Gonder and Bahir Dar in the Ethiopian plateau with long records were used (Fig. 1).

Daily streamflow records for 41 years (1972-2011) at two hydrological stations (Al-Gewisi and Al-Hawata) on the Dinder and Rahad Rivers respectively, were obtained from the Ministry of Water Resources, Irrigation and Electricity-Sudan (MoWRIE). The daily data were used for analysing the streamflow parameters of magnitude, timing, frequency, duration and rate of change of flows using the IHA approach. Monthly mean, maximum, mean annual and maximum annual streamflow were calculated for the trends analysis. Gedarif annual maximum and minimum temperature for the period (1941-2008), and the long term monthly mean, maximum and minimum temperature records for the period (1941-2010), were obtained from the Sudanese Meteorology Authority. Temperature data for Gonder station close to the upper part of the D&R, was obtained from the Ethiopian National Meteorological Agency. Figure 3 shows the annual minimum and maximum temperature for Gedarif and Gonder stations. The hydro-climatic data in the D&R, is generally scant with many data gapsand may contain measurements and/or typos errors. Abnormal values and outliers could lead to wrong conclusion. Therefore, removing such errors is critical in data mining and data analysis, especially when analysing trends. Thus, we carefully examined all the data before analysing trends. Data screening and data quality checks were performed for all data sets before analysis. Visual inspection and regression analysis between neighbouring stations were used to identify outliers and fill in missing data in the data sets. For instance we found that temperature data is accurate, while streamflow data contained outliers and typos errors, which were corrected. Continuous missing data for a length of one year and above, were omitted from the analysis.

Due to data scarcity in the region, some of the climate data were obtained from neighbouring stations outside the case study boundary, but within the same climate zone. We analysed the reference evapotranspiration (ET$_0$) in the region to support the analysis of using neighbouring stations. Long-term monthly ET$_0$ data for some of the examined stations inside and outside the case study boundary was obtained from IWMI Online Climate Summary Service Portal (http://wcatlas.iwmi.org/results.asp). The analysis has shown that the ET$_0$ for the examined neighbouring stations (i.e. Gonder, Bahir Dar, Ad Damazin, Abu Naama





and Sennar) have similar patterns to those stations inside the case study boundary (i.e. Al-Hawata). Figure 2 shows the long-term monthly $ET_0$ for some of the examined stations in the region.

# 4    Results and discussion

This section presents the results of the statistical tests to assess long term trends of the D&R hydro-climatology. To support the statistical test and to have critical analysis of streamflow, alteration in streamflow through IHA is also discussed.

## 4.1    MK and Pittitt analysis

### 4.1.1    Precipitation

For the 12 gauging stations of precipitation (Fig. 1), the MK test shows no long term significant trends at 5% confidence

level, over the D&R basin. Only one station (Doka) shows a significant increasing trend with values of 0.332 and 0.006 for $\tau$ and $p$, respectively. These results agree with literature on precipitation trends over the neighbouring basin of the Blue Nile. For example Tesemma et al. (2010) showed no change in precipitation in the Blue Nile basin during (1963-2003). Gebremichael et al. (2013), investigating trends in precipitation in the Blue Nile with records between the 1970s and the beginning of the 21st century, found no significant change in the annual precipitation in the upper Blue Nile basin. Tekleab et

al. (2013), applied the statistical MK test to study the trends in rainfall, temperature and streamflow in the Abbay/Upper Blue Nile basin. His result found not significant trends in precipitation in all inspected stations.

Those studies reported no significant trends in precipitation across the Abbay/Upper Blue Nile basin, which includes the upper D&R basin. The MK results were found to be sensitive to the time domain.

### 4.1.2    Temperature

Unfortunately, only Gedarif station have a long record of temperature data. Gedarif may be considered representative for the downstream part of the D&R basin (Fig. 1). Temperature data for this station was analyzed for the period (1941-2010). Since there is no temperature data in the upper part of the D&R basin, the neighbouring station of Gonder was considered representative for the upper part of the basin (Fig. 1). The MK tests results are shown in Table 2.

As expected, both Gonder and Gedarif show a significant increasing trend of temperature, with mean annual temperature

increasing at 0.03 °C/y in Gondar, and at 0.02 °C/y in Gedarif. It is expected that increased temperature, particularly during the dry season (November -June), may influence evapotranspiration from the mayas, leading to increased dryness.

The Global Climate Model (GCM) results in the upper Blue Nlie Basin showed both increasing and decreasing trends in precipitation, but agreed on a temperature rises (Elshamy et al. 2009; Nawaz et al., 2010).



### 4.1.3    Streamflow

Mean annual flow showed a significant increasing trend for the Rahad River at Al-Hawata station, but not for the Dinder River at Al-Gewisi station. While, the annual maximum flow of the Dinder showed significant decreasing trend, it is not the case for Rahad (Table 3). The statistical tests of the seasonal time series of Rahad showed significant increasing trends of the monthly

mean for July, August and November. While the monthly maximum flows showed significant decreasing trend in August flow and increasing trend in November flow of the Dinder River at Al-Gewisi station, there was no evidence for significant trend for the Rahad River at Al-Hawata station (Table 3). Since August is the period of high flow in both the Dinder and the Rahad rivers, increasing flow in this period leads to inundation of floodplain including mayas, while decreasing flow leads to dryness of mayas. Figure 4 showed the Pettitt tests results for the abrupt changing points. Detailed analysis of IHA for the five

environmental flow components (magnitude, frequency, duration, timing and rate of change of flow) are discussed below. Since Dinder River supports the ecosystem of the DNP floodplain (Mayas), our IHA analysis focused on the alterations of high extremes parameters.

The importance of large floods (flows equal to or greater than the 10-year return period flood) is to inundate the Dinder River floodplain wetlands (Mayas). Therefore, we hypothesize that alterations in the magnitude, frequency, timing and duration of

the annual large flood peaks have affect production of native river-floodplain flora and fauna. The small flood pulse (flows equal to or greater than bankfull flows but less than the 10-year return period flood) inundates the mayas wetlands to a shallower depth, with the result that forage and water for wildlife remained available for only a short period of time.

### 4.2    IHA analysis

### 4.2.1    Magnitude of monthly flow

For IHA analysis, the record of time series is divided into two parts: pre-impact (1972-1991), and post-impact (1992-2011). The general pattern of median monthly flow of the Dinder River at Al-Gewisi station is that the median flow increased in July and November at the beginning and end of the rainy season (period of low flow), and decreased in August and October ( Period of high flow). The median monthly flows of July and November increased from 43 and 0 (dry) $m^3/s$ to 50 and 14 $m^3/s$, respectively. In contrast, the median monthly flows of August and October decreased by 20% and 11% from 266 and 101 $m^3/s$

to 210 and 90 $m^3/s$, respectively.

In comparison to Dinder and similarly to MK test, the Rahad median monthly flows showed increasing patterns in all months, with increasing pattern from 45, 133 and 0 $m^3/s$ to 65, 153 and 14 $m^3/s$ in July, August and November, respectively. The monthly flows are shown in (Fig. 5). The alteration of the monthly flow magnitude between pre and post-impact periods in particular during months of high flows (August-October) is likely affect habitat availability in particular on floodplains, which

may lead to decrease or even disappearance of native plants species and increase in non-natives plants species that might not be suitable for the herbivores wildlife that inhabit the DNP.



### 4.2.2    Magnitude and duration of river extreme floods

Extreme floods are important in re-forming both the physical and biological structure of a river and its associated floodplains such as oxbow lakes and wetlands.  For the mayas wetlands of the DNP, all results show a decreasing maxima trend for the Dinder River. The post-impact median flow maxima for 1, 7, 30 and 90-day intervals were, 14 %, 13%, 15%, and 14%, lower than pre-impact. In contrast, in the Rahad River increasing patterns were observed, with a post-impact median flow maxima for 1, 7, 30 and 90-day of 6 %, 9%, 16%, and 21%, respectively higher than pre-impact (Fig. 6). Peak flows are critical aspect of the lateral connectivity between Dinder and Rahad rivers and its associated floodplains (mayas). The alterations in the Dinder river flow are likely affect the ecosystems in DNP negatively. Decrease in magnitude and duration of the annual flood peaks that reduce the amount of water flowing to the river-floodplain, may diminish the production of native flora and fauna, and animal migration that may be linked to floodplain inundation.

### 4.2.3    Timing of annual extreme floods

In the Dinder River, timing of the annual maximum daily flow before and after flow impact happened within the same two weeks (16 September – 02 September, Julian date (JD) 260–246), but 14 days earlier. The large flood peak flows occurred twice during the pre-impact period. The first peak occurred on the 16th of September 1975, with flow peak reached 1010 m$^3$/s. The second peak occurred on the 2nd of September 1988, with flow peak reached 834 m$^3$/s. On the other hand the post-impact period showed zero large flood peaks.  In the Rahad River, timing of the annual maximum daily flows before and after flow impact happened also within the same two weeks (22 September – 10 September, Julian date (JD) 266–254), but 12 days earlier. The large flood peak flows occurred twice during the pre-impact period. The first peak occurred on the 9th of October 1974 with flow peak reached 190 m$^3$/s. The second peak occurred on the 5th of September 1981 with flow peak reached 206 m$^3$/s. On the other hand the post-impact period showed more frequency of large flood peaks.   Four large flood peaks were occurred during this period; on the 17th of September 1994, the 2nd of September 2007, the 27th of August 2008 and the 20th of September 2010, with peak flows of 192, 196, 201, and 201 m$^3$/s, respectively. Synchronization of annual flood with life cycle requirements of a range of riverine and floodplain species is of likely high importance given the adaptation of species to their habitat. Timing shift of the Dinder river peak flow may leads to desynchronization with the life cycle requirements of some of the species.

### 4.2.4    Rate and frequency of change in flow

Median rate of flow rises (positive differences between consecutive daily values) in the Dinder River has decreased by 38% from 32 m$^3$/s/day during the pre-impact period to 20 m$^3$/s/day during the post-impact period. The median rate of flow falls



(negative differences between consecutive daily values) has decreased by 53% from 17 m$^3$/s/day during the pre-impact period to 8 m$^3$/s/day during the post-impact period and the median number of flow reversals per year has increased from 39 to 43 m$^3$/s/day. Likewise, the median rate of flow rises in Rahad River has decreased by 40% from 5 m$^3$/s/day during the pre-impact period to 3 m$^3$/s/day during the post-impact period. The median rate of flow falls has decreased by 60% from 5 m$^3$/s/day during

the pre-impact period to 2 m$^3$/s/day during the post-impact period and the median number of flow reversals per year decreased by 6% during the post-impact period from 33 m$^3$/s/day to 31 m$^3$/s/day (Fig. 7). The rate of change in flow can affect persistence and life time for both aquatic and riparian species (Poff et al., 1997), particulary in such arid area where streamflow can change rapidly in s very short period of time due to excessive rainfall.

### 4.2.5     The Hydrological Alterations factors with RVA categories

The Dinder streamflow Hydrologic Alteration factors are reflected in Fig. 8 (a). Monthly flows from July to October show that the post-impact flows have decreased in the high RVA category and increased in the middle and low RVA categories (i.e. the post-impact period of record has less than the expected number of years in the high RVA category based on the pre-impact flows). In contrast, the monthly flow of November has increased in the high RVA category and decreased in the middle RVA category. The annual flow maxima of 1, 7, 30 and 90-day have decreased in the high and middle RVA categories and increased

in the low RVA category. Conversely, the number of zero flow days has decreased in the high and middle RVA categories and increased in the low RVA category, while the date of maximum flow has increased in the high RVA category and decreased in the middle RVA category. The high pulses count has increased in the high RVA category and decreased in the middle RVA category, while the high pulses duration has decreased in the high and middle RVA categories and increased in the low RVA category. The rate of rising flow has decreased in the high and middle RVA categories and increased in the low RVA category.

In contrast, rate of falling flow has increased in high RVA category and decreased in middle and low RVA categories. Similar to falls rate, the number of reversals has increased in high RVA category and decreased in middle and low RVA categories.

The Rahad streamflow Hydrological Alterations factors are reflected in Fig. 8 (b). Monthly flows from June to November show that the post-impact flows have increased in the high RVA category and decreased in the middle and low RVA categories (except August which has increased in both high and middle categories and decreased in low category). The annual flow

maxima of 1, 7, 30 and 90-day have increased in the high and middle RVA categories and decreased in the low RVA category. The number of zero flow days has decreased in the high and middle RVA categories and increased in the low RVA category, while the date of maximum flow has increased in the high RVA category and decreased in the middle and low RVA categories. The high pulses count and duration have slightly increased in the high and middle RVA categories and decreased in the low RVA category. The rise rate has decreased in the high and middle RVA categories and increased in the low RVA category. In

contrast, the falls rate has increased in high RVA category and decreased in middle and low RVA categories. While the number of reversals has increased in high and low RVA categories and decreased in middle RVA category.



Hydrologic alterations with value equal to zero indicate that the observed frequency of post-impact annual parameter values fell within the RVA category equals the expected frequency. A positive value indicates that annual parameter values fell inside the RVA target category more often than expected, while negative values indicate that annual values fell within the RVA target category less often than expected.

## 5    Conclusions

The long term trend of the Dinder and Rahad hydro-climatology has been analysed for twelve precipitation, one temperature, and two streamflow gauging stations, over different periods of time. The MK test showed statistically significant increasing trends in temperature, but no significant changes in precipitation. The trend results on precipitations agree with the literature on neighbouring catchments of the Blue Nile (e.g.Tesemma et al., 2010; Gebremichael et al., 2013; Tekleab et al., 2013).

The mean annual streamflow of the Rahad River exhibited statistically significant increasing trend, but not for the Dinder River which showed no significant changes. The trend of the monthly mean flows showed significant increasing trends in Rahad River for July, August and November, while no significant trend was observed in Dinder River. The monthly maxima flows showed a significantly decreasing trend for August maxima flows and decreasing trend for November maxima flows in the Dinder River, while no evidence for significant trend in monthly maxima flows of the Rahad River. Increasing trends in Rahad flows could be attributed to LULC change in the upper catchment that accelerate the runoff response. The upper catchment (effective catchment where rainfall is relatively high) of the Rahad River is small compared to upper Dinder. Therefore, same magnitude of LULC changes is expected to have more impact on streamflow in Rahad than in Dinder. This is likely the reason for the significant increasing trends for Rahad flows. Reduction of the Dinder peak flow can have direct impact on filling of the mayas wetland, the main water source for the DNP during the dry months. The results of increasing temperature associated with increasing flow in Rahad River, indicate that increasing trend of temperature shall not always lead to decreasing discharge as LULC change is another important factor in the partitioning of precipitation.

The IHA analysis has shown that the flow of the Rahad River was associated with significant upward alterations in some of the hydrological indicators. The Flow of the Dinder River was associated with significant downward alterations. Particularly, these were: a) a decrease in the magnitude of the river flow during August (peak flow), and an increase of low flows (November); b) a decrease in magnitude and duration of flow extremes (i.e. 1,7, 30 and 90-day maxima); c) a decrease in the frequency of annual flow extremes of large flood peaks; d) an increase in duration and frequency of annual high-flow pulses; e) an increase in flow reversals frequency combined with a reduction in rises rate and increase in falls rate in streamflow. This alterations in the Dinder river flow are likely affect the ecosystems in DNP negatively. The importance of annual floods that inundate the Dinder River floodplain wetlands (mayas) are likely to have significant effects on a range of species that depend on seasonal patterns of flow. Therefore, alterations in magnitude of the annual large flood peaks that reduce the amount of water flowing to the mayas, may diminish the production of native river-floodplain flora and fauna, and animal migration that may be linked to wetlands inundation.



The result of no significant increasing/decreasing trends of precipitation over D&R basin, indicates other factors than climate variability might be responsible for streamflow alteration. This could be attributed to large scale land use change that accelerate the runoff response. LULC changes in neighbouring catchments of the Blue Nile headwaters has been widely reported in literature (e.g. Zeleke and Hurni, 2001; Tesemma et al., 2010; Teferi et al., 2010; Gumindoga et al., 2014). How far this caused modified water fluxes downstream will be part of future investigations.

*Competing interests*: The third author is a member of the editorial board of the journal.

*Acknowledgements*. This study was carried out as part of a PhD research programme of the first author entitled "The impacts of land degradation on the Dinder and Rahad hydrology and morphology, and linkage to the ecohydrological system of the Dinder National Park, Sudan'', which is funded by the Netherland Fellowship Programme (NFP). The authors would like to extend their appreciation to the Hydraulics Research Center of Ministry of Water Resources, Irrigation and Electricity-Sudan for providing the data.



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


Table 1: Available precipitation monthly data in the study area (mm/y)

| Station | Name | Data availability | $P_{max}$ | $P_{min}$ | $P_{mean}$ |
|---|---|---|---|---|---|
| 1 | Gedarif | 1903-2012 | 1035.3 | 289 | 608 |
| 2 | Gadambaleya | 1979-2012 | 779.6 | 303 | 544 |
| 3 | Samsam | 1979-2012 | 886 | 427 | 701 |
| 4 | Um seinat | 1979-2012 | 1070 | 379 | 665 |
| 5 | Doka | 1979-2012 | 1016 | 414 | 682 |
| 6 | Al-Hawata | 1979-2012 | 917 | 222 | 511 |
| 7 | Sennar | 1907-2008 | 758 | 175 | 440 |
| 8 | Ad Damazin | 1981-2000 | 899 | 497 | 700 |
| 9 | Abu Naama | 1984-1998 | 815 | 372 | 606 |
| 10 | Um Benien | 1984-1998 | 715 | 313 | 507 |
| 11 | Gonder | 1953-2007 | 1823 | 720 | 1117 |
| 12 | Bahir Dar | 1961-2007 | 2036.7 | 894.5 | 1420 |

*Source*: (Sudanese Meteorology Authority and Global Historical Climatology Network (GHCN))



Table 2: Mann-Kendall results for Gonder and Gedarif mean annual temperature

| Station | Kendall's $\tau$ | S | P-value | Trend |
|---------|------------------|-----|---------|-------|
| Gonder | 0.3227 | 111 | 0.02118 | Significantly increasing |
| Gedarif | 0.3126 | 755 | 0.00010 | Significantly increasing |

S    : The (Kendall) S-statistic value

5    $\tau$ :The Kendall rank-correlation coefficient ($\tau$ )

p    :The p-value (computed probability)

30





Table 3. Mann-Kendall tests results of annual and seasonal flow for Dinder at Al-Gewisi and Rahad at Al-Hawata at 5% confidence level (P = 0.05)

| | River | Kendall's τ | S | P-value | Trend |
|---|---|---|---|---|---|
| Mean annual | Dinder | -0.146 | -114 | 0.189 | No significant trend |
| | Rahad | 0.256 | 200 | **0.020** | Significantly increasing |
| Annual maxima | Dinder | -0.280 | -0.220 | **0.010** | Significantly decreasing |
| | Rahad | 0.163 | 127 | 0.142 | No significant trend |
| August maxima | Dinder | -0.338 | -0.264 | **0.002** | Significantly decreasing |
| | Rahad | 0.195 | 152 | 0.079 | No significant trend |
| November maxima | Dinder | 0.232 | 174 | **0.041** | Significantly increasing |
| | Rahad | 0.170 | 130 | 0.130 | No significant trend |
| July mean | Dinder | 0.153 | -120 | 0.165 | No significant trend |
| | Rahad | 0.232 | 181 | **0.036** | Significantly increasing |
| August mean | Dinder | -0.210 | -164 | 0.057 | No significant trend |
| | Rahad | 0.256 | 200 | **0.020** | Significantly increasing |
| November mean | Dinder | 0.315 | 237 | **0.005** | Significantly increasing |
| | Rahad | 0.255 | 194 | **0.024** | Significantly increasing |





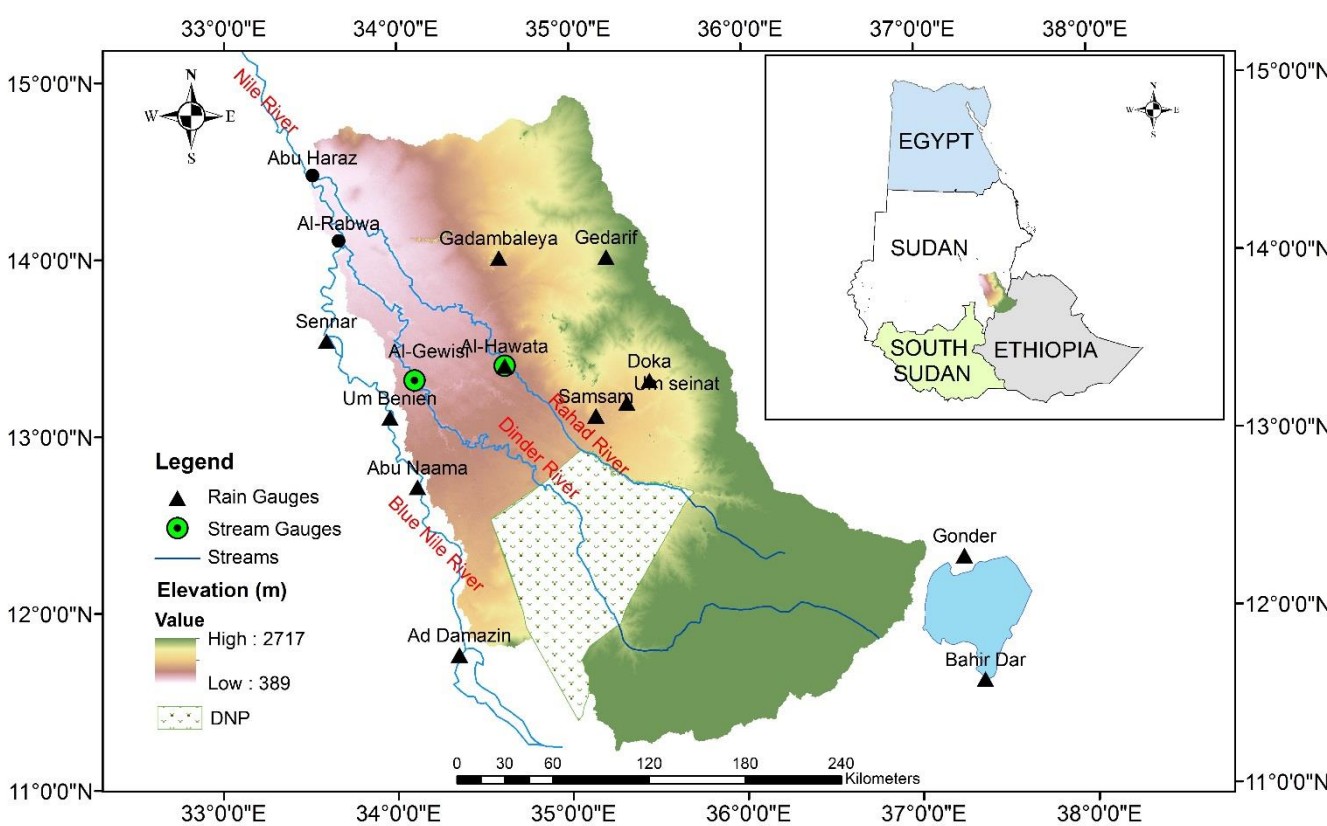

Figure 1: Locations of the D&R basin and the hydro meteorological stations used in this study

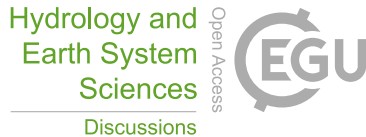

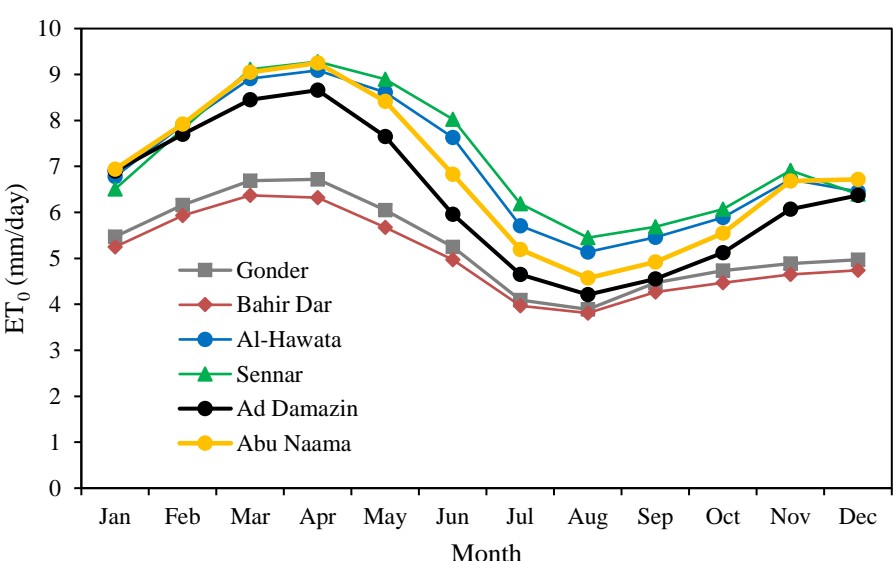

Figure 2: $ET_0$ for the examined stations (Gonder, Bahir Dar, Ad Damazin, Abu Naama and Sennar) outside the case study boundary, and (Al-Hawata) stations inside the case study boundary.





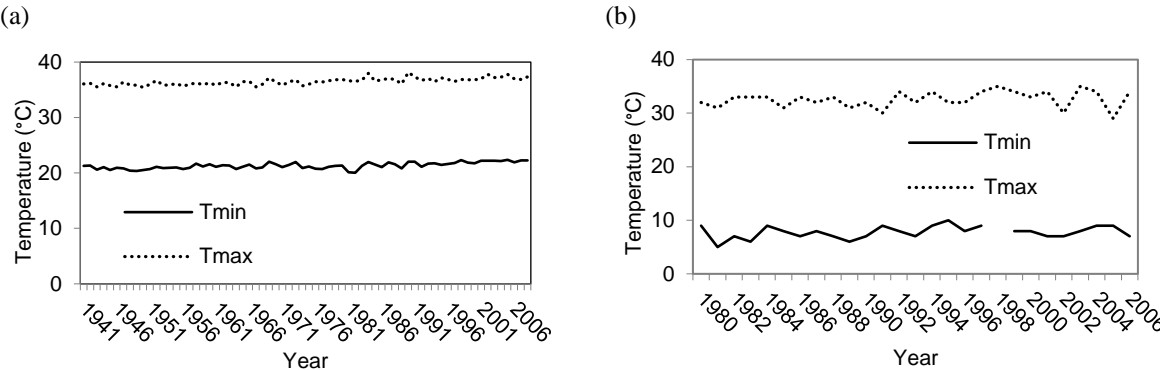

Figure 3: Annual minimum and maximum temperature: (a) for Gedarif station and (b) for Gonder station




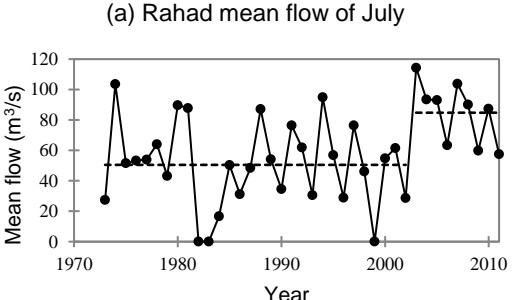

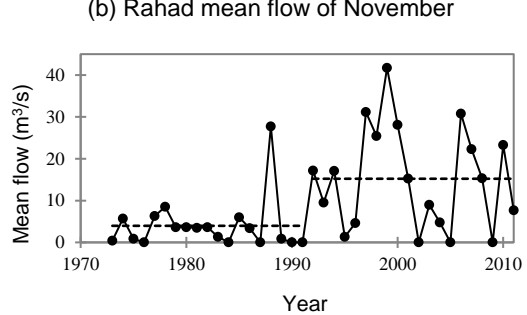

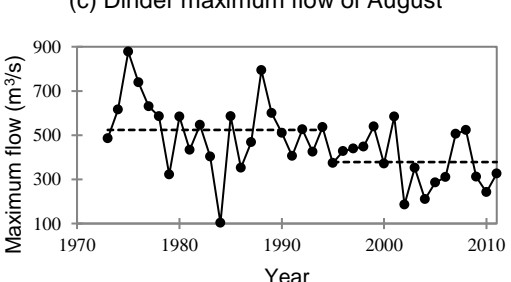

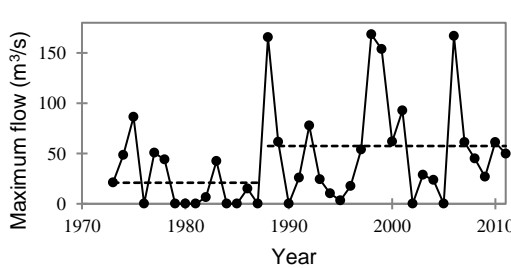

5    Figure 4: The Pettitt homogeneity test for detecting the abrupt changing points of seasonal flows for (a) and (b) Rahad River, and (c) and (d) Dinder River. The dash lines are the mean of the time series before and after the change point.





Fig. 5: Seasonal flow pattern for (a) Dinder River and (b) Rahad River



(a)

(b)

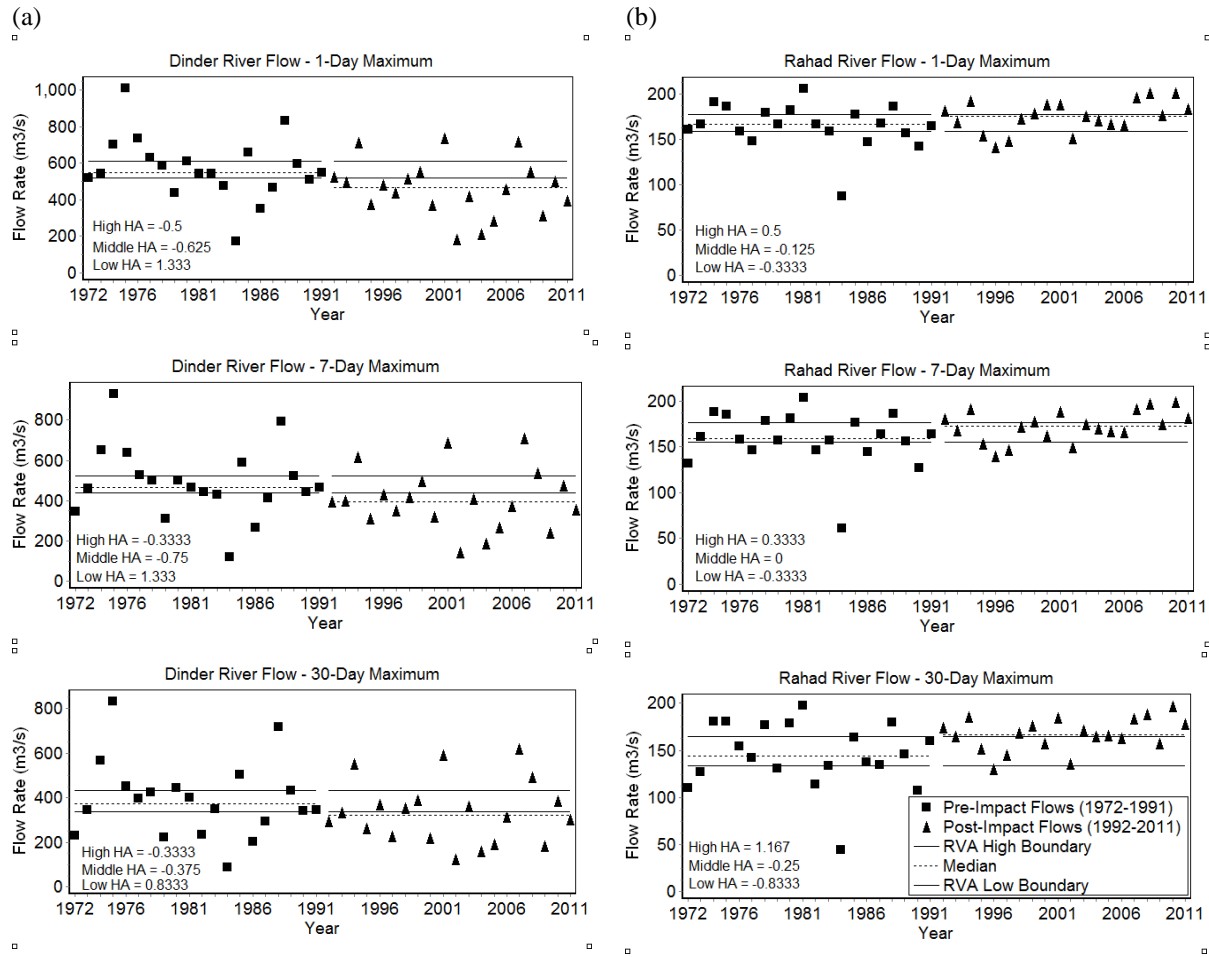

Figure 6: Annual 1, 7, and 30-day maximum flows for (a) Dinder River and (b) Rahad River



(a)

(b)

Figure 7: Rates of flow rises and falls and numbers of reversals for (a) Dinder River and (b) Rahad River



(a)



(b)

Figure 8: shows the Hydrological Alterations factors for all parameters except for Environmental Flow Components (EFCs) for (a) Dinder streamflow and (b) Rahad streamflow. RVA analysis is not available for EFCs.