# Peer review of "Investigation of the long-term variations in hydro-climatology of the Dinder and Rahad basins and its implications on ecosystems of the Dinder National Park, Sudan"

_Hydrology and Earth System Sciences, 2016_

## Referee Comment (RC1) · W. Mbungu (Referee) · 7 Nov 2016

This paper is generally of good quality, as methods used in the analysis are acceptable and have been widely used in other areas for trend analysis and for understanding the hydrologic alterations. The authors have elaborative, except for the few areas that have been outlined below. The presentation has been good and it is a contribution to knowledge Variability of hydro-climatic parameters has a great influence on the hydrology and sustainability of ecosystems in many landscapes around the world. The situation is alarming in Africa, because of the anthropogenic influence. In order to properly man-

age water resources, knowledge of variability and its impacts on hydrology is essential. This study has played a great role in understanding trends in rainfall, temperature and streamflow. The use of temperature as a proxy for evapotranspiration is especially interesting as it indicates the increasing or decreasing of evapotranspiration which has a great influence on the hydrology of the landscape or watershed. The use of the trend test is especially useful for a snapshot view of the general direction and magnitude of trends. In addition, the use of the using the Indicators of Hydrologic Alterations (IHA) has been important for understanding stream modifications in relation to ecological functions.

In line 22-25, the authors describe that there were missing gaps of the data variables, and had used visual inspection and regression analysis to fill in the missing gaps, and further describe about the streamflow data that had typos and outliers, but did not elaborate if the methods used were common for all the data variables or were specific. It will be especially important to note how missing gaps in flows were treated, for which I am sure are not going to be the same as rainfall data which can utilize the neighboring stations. Line 30... describes about the availability of long-term ETo data for some areas in the watershed, I am surprised why the authors did not want to use the data in the trend analysis. Section 4.1.1: line 9 describes that there are no significant trends at 5% level of significance from the 12 precipitation stations, but we do not see the data that support that. It is only a statement describing that, but it would be useful to have data that confirms what they describe.

Some Typos and other errors Section 4.1.2: line 22-23: these can be moved to the methodology section Section 4.2.1: line 29: ...."is likely affect".... Should read ..."is likely to affect".... Section 4.2.2: line 9........"are likely affect" ...... should read ...."are likely to affect"...... Section 4.2.3: line 23-24 ...."Four large flood peaks were occurred ......" should read "....four large flood peaks occurred....." Section 4.2.3 line 26 ...."...Timing shift of the Dinder river peak flow may leads to...." Should read "timing shift of the Dinder river peak flow may lead to..." Section 4.2.4: page 12 line 3-4:

the sentence "Likewise, the median rate of flow rises in Rahad River has decreased by 40% from 5 m3/s/day during the pre-impact period to 3 m3/s/day during the post-impact period." The sentence needs to be re-written to avoid ambiguity. Line 15: the phrase "His result found not significant trends in precipitation in all inspected stations" needs to be re-written as it is difficult to understand what the his refers to... Line 17 should read: LULC changes have widely been reported..... Line 21 the words "gapsand" should read "gaps and" End of line 22- beginning of line 23 ...... has been reported in the literature......

---

## Referee Comment (RC2) · Anonymous Referee #2 · 27 Nov 2016

The topic is interesting and its content is useful for research and for development as well with clear objective of investigating the long-term variations of stream flow, rainfall and temperature over the D&R and its implications on the DNP wetlands ecosystems. However, the followings are my general comments

1. LULC changes occurred around early 1990's in the upper Dinder and Rahad was the assumption for the study and 1992 was the separating time period for the analysis of the changes for the IHA statistical comparison. However, this assumption can be proved or analyzed by LULC change detection techniques.

2. The IHA technique computes 33 hydrologic parameters for each 10 year. But parameters and their ecosystem influences were not mentioned anywhere (figure 8) and also how parameters are calculated. Example, Mean or median value will tell us about the reliability of water for aquatic animals living in Mayas.

3. Observed and expected frequency were used to calculate hydrological alteration factor (HA) but it is not well described what is expected frequency and how to find or calculate expected frequency.

4. They used 12 stations for precipitation, 1 station for temperature and 2 flow stations. In my opinion 1 temperature station will not be enough to represent spatial variability of the catchments. Also, Gondar and Bahirdar cannot represent the two catchments as they are found at the highlands of UBNRB. Moreover, they used regression analysis between neighboring stations to fill in missing data, which is not clear how they fill the gaps of a single temperature station. There is no information how they filled the missing values of streamflow or how they detct outliers.

5. The IHA analysis indicated that the Rahad River flow was coupled with significant upward alterations in some of the hydrological indicators. In contrast, the Dinder River flow was coupled with significant downward alterations. Alterations in magnitude and duration of the annual flood peaks that reduce the amount of water flowing to the river-floodplain, may diminish the production of native flora and fauna, and animal migration that may be linked to floodplain inundation. This conclusion is too general as there is no any standard set for habitat suitability indices to quantify what percentage of flow variation affects the flora and fauna?

6. There is no any drawn conclusion from RVA analysis and also from change detected of precipitation and temperature. The rise in temperature may not or may be favorable for animals living in DNP......

7. Figures are not well described or clarified in text. For example figure 4, 5 and 6 showed peaks or abrupt change points but nothing was mentioned what was happened

during that time on the Mayas ecosystem or the historical impacts of the high and low flow. In general it lacks detail description of methodology and summarized result analysis and conclusion of the 33 parameters.

Some additional comments to specific paragraphs:

a) page 4, line 25-30: The ESS should be clearly distinguished. Seperate them into the common categories, also by using a table.

b) page 5, line 21-23: Which loss of biodiversity occured? Give facts!

c) page 7, line 30-32: Which kind of significance tests are used for the HA?

d) figure 2: Which ET0 formula was used? How sensitive are the absolute ET0 values?

e) figures 5 -7: The significance of the decrease / increase is not explained or tested. Moreover, there are some 0-values which are probably "no data" values (e.g. Rahad River in August).

f) page 12, line 10-31: In this form not readable (you get "dizzy").

g)page 13, line 15-18: Quantify the LULC!

---

## Author Comment (AC1) · 25 Dec 2016

The Authors highly appreciate the suggestions and constructive criticisms posed by the reviewers. We also would like to thank the editor Dr. Uwe Ehret for handling the review process of the manuscript. Here we present our response to the discussion issues that have been arisen during the review process.

1- Referee #1:

The Authors would like to thank the referee Dr.Winfred Mbungu for the time dedicated to review this manuscript. We greatly value his comments and suggestions; they helped

us to improve the manuscript. We have given feedback after each comment as follows:

General comments: This paper is generally of good quality, as methods used in the analysis are acceptable and have been widely used in other areas for trend analysis and for understanding the hydrologic alterations. The authors have elaborative, except for the few areas that have been outlined below. The presentation has been good and it is a contribution to knowledge Variability of hydro-climatic parameters has a great influence on the hydrology and sustainability of ecosystems in many landscapes around the world. The situation is alarming in Africa, because of the anthropogenic influence. In order to properly manage water resources, knowledge of variability and its impacts on hydrology is essential. This study has played a great role in understanding trends in rainfall, temperature and streamflow. The use of temperature as a proxy for evapotranspiration is especially interesting as it indicates the increasing or decreasing of evapotranspiration which has a great influence on the hydrology of the landscape or watershed. The use of the trend test is especially useful for a snapshot view of the general direction and magnitude of trends. In addition, the use of the using the Indicators of Hydrologic Alterations (IHA) has been important for understanding stream modifications in relation to ecological functions.

Comment: In line 22-25, the authors describe that there were missing gaps of the data variables, and had used visual inspection and regression analysis to fill in the missing gaps, and further describe about the streamflow data that had typos and outliers, but did not elaborate if the methods used were common for all the data variables or were specific. It will be especially important to note how missing gaps in flows were treated, for which Iam sure are not going to be the same as rainfall data which can utilize the neighboring stations.

Response: Different methods have been used to fill in the missing gaps of the data variables. Regression analysis was used to fill in the missing gaps in monthly precipitation. Continuous missing data for a length of one year were omitted from the analysis. For flow data, missing data of a short duration (e.g. 1-2 days) was filled using linear interpolations. Missing data of more than 2 days was filled using rating curves. However, missing flow data for a length of one month or more were omitted from the analysis. This paragraph was added to section 3.3.

Comment: Page 8, line 30: describes about the availability of long-term ETo data for some areas in the watershed, I am surprised why the authors did not want to use the data in the trend analysis.

Response: Unfortunately, what is available is the long term mean values of ET0 from IWMI Online Climate Summary Service Portal (http://wcatlas.iwmi.org/results.asp) and not the time series itself. ET0 was used here to check whether climate data are from similar climate zones.

Comment: Section 4.1.1: line 9 describes that there are no significant trends at 5% level of significance from the 12 precipitation stations, but we do not see the data that support that. It is only a statement describing that, but it would be useful to have data that confirms what they describe.

Response: Table 3 showing the Man-Kendall results of the 12 precipitation stations is added to the manuscript. Comment: Some Typos and other errors Section 4.1.2: line 22-23: these can be moved to the methodology section

Response: Agreed. Line 20-23 were moved to the methodology section.

Comment: Section 4.2.1: line 29 "is likely affect" Should read "is likely to affect"

Response: Corrected to "is likely to affect"

Comment: Section 4.2.2: line 9 "are likely affect" should read "are likely to affect"

Response: Corrected to "are likely to affect"

Comment: Section 4.2.3: line 23-24: "Four large flood peaks were occurred" should read "four large flood peaks occurred"

Response: Corrected to "four large flood peaks occurred"

Comment: Section 4.2.3 line 26 "Timing shift of the Dinder river peak flow may leads to " Should read "timing shift of the Dinder river peak flow may lead to"

Response: Corrected to "timing shift of the Dinder river peak flow may lead to".

Comment: Section 4.2.4: page 12 line 3-4: the sentence "Likewise, the median rate of flow rises in Rahad River has decreased by 40% from 5 m3/s/day during the pre-impact period to 3 m3/s/day during the post-impact period." The sentence needs to be re-written to avoid ambiguity.

Response: The sentence was rewritten as: "Similar to Dinder, the median rate of flow rises in Rahad River has decreased by 40% from 5 m3/s/day during the pre-impact period to 3 m3/s/day during the post-impact period."

Comment: Page 9, Line 15-16: the phrase "His result found not significant trends in precipitation in all inspected stations" needs to be re-written as it is difficult to understand what the his refers to.

Response: sentence was rewritten to: "Using the MK test, Tekleab et al. (2013) found no significant trends of precipitation in the Abbay/Upper Blue Nile basin"

Comment: Page 3, Line 17 should read: LULC changes have widely been reported

Response: The sentence was corrected to "LULC changes were reported widely". Same correction on Page 14, line 2-3: "LULC changes in neighbouring catchments of the Blue Nile headwaters has been widely reported in literature" was corrected to "LULC changes in neighbouring catchments of the Blue Nile headwaters have widely been reported in the literature"

Comment: Page 8, Line 21 the words "gapsand" should read "gaps and"

Response: Corrected to "gaps and"

Comment: Page 3, End of line 22- beginning of line 23: has been reported in the literature

Response: Corrected to "has been reported in the literature".

2- Anonymous Referee #2:

We thank the anonymous referee #2 for his/her constructive comments. His/her suggestions and comments helped us to improve the quality and readability of the manuscript. We have responded to all comments as given below: General comments: The topic is interesting and its content is useful for research and for development as well with clear objective of investigating the long-term variations of stream flow, rainfall and temperature over the D&R and its implications on the DNP wetlands ecosystems. However, the followings are my general comments:

1. LULC changes occurred around early 1990's in the upper Dinder and Rahad was the assumption for the study and 1992 was the separating time period for the analysis of the changes for the IHA statistical comparison. However, this assumption can be proved or analyzed by LULC change detection techniques.

Response: This is correct, i.e., to use LULC changes detection technique to investigate alteration in hydrology. In fact, this is the content of our next paper. Here, the hypothesis is to investigate if climate, i.e., trends of temperature and precipitation are responsible for the alteration in hydrology. It becomes unacceptably long manuscript to combine both derivers (climate and LULC changes), responsible for hydrological alteration.

2. The IHA technique computes 33 hydrologic parameters for each 10 year. But parameters and their ecosystem influences were not mentioned anywhere (figure 8) and also how parameters are calculated. Example, Mean or median value will tell us about the reliability of water for aquatic animals living in Mayas.

The referee is correct, only general reference is made to the effect of hydrology alteration to ecosystem, e.g., as given in Section 4.2.1: "The alteration of the monthly

flow magnitude between pre and post-impact periods in particular during months of high flows (August-October) is likely to affect habitat availability in particular on flood-plains, which may lead to decrease or even disappearance of native plants species and increase in non-natives plants species that might not be suitable for the herbivores wildlife that inhabit the DNP" and also in Section 4.2.2. However, the detailed analysis of hydrological influence on the ecosystem is beyond the scope of this paper, otherwise the paper becomes too long.

3. Observed and expected frequency were used to calculate hydrological alteration factor (HA) but it is not well described what is expected frequency and how to find or calculate expected frequency.

Response: In an RVA analysis, the full range of pre-impact data for each parameter is divided into three different categories. The default method for non-parametric analysis is to divide the data into three equal categories (0-33rd percentile, 34th-67th percentile, and 68th-100th percentile). The IHA software next computes the expected frequency with which the "post-impact" values of the IHA parameter should fall within each category, based on the pre-impact frequencies (in the non-parametric default, this would be 33% of the annual values in each of the three categories). Then it computes the frequency with which the "post-impact" annual values of IHA parameters actually fell within each of the three categories. A Hydrologic Alteration factor is then calculated for each of the three categories as shown in Eq.7 in the main manuscript. More details can be found in the IHA V7.1 Tutorial prepared by The Nature of Conservancy (2009). A paragraph describing how expected frequency was calculated, was added to the methodology section of the main manuscript.

4. They used 12 stations for precipitation, 1 station for temperature and 2 flow stations. In my opinion 1 temperature station will not be enough to represent spatial variability of the catchments. Also, Gondar and Bahir dar cannot represent the two catchments as they are found at the highlands of UBNRB. Moreover, they used regression analysis between neighboring stations to fill in missing data, which is not clear how they fill the

gaps of a single temperature station. There is no information how they filled the missing values of streamflow or how they detect outliers.

Response: The referee is right that, data availability is really a challenge in this region. However, the long term rise of temperature has widely been reported in the literature (e.g. Tekleab et al. 2013, Bashir et al. 2016),), while the results of precipitation trends has been controversial. This is why our focus was more on long term rainfall data. We have analysed 2 temperature stations with long records, one to represent the highland (Gondar), and the second for the low land ((Gedarif). The results also agreed with literature, in that there is a temperature rise of between 0.2 to 0.6 oC/decade during the last decades.

For the rainfall stations, we assumed Gonder and Bahir Dar to represent the highland of the D&R, while the remaining stations to represent the low land of the catchment.

We have used reference crop evapotranspiration ET0 as a proxy to verify similarity of climate zones before using neighbouring stations in regression analysis. Furthermore, the regression is rejected if it shows low regression coefficient (R2 < 0.5). This is added in page 8, lines 29-30.

5. The IHA analysis indicated that the Rahad River flow was coupled with significant upward alterations in some of the hydrological indicators. In contrast, the Dinder River flow was coupled with significant downward alterations. Alterations in magnitude and duration of the annual flood peaks that reduce the amount of water flowing to the river floodplain, may diminish the production of native flora and fauna, and animal migration that may be linked to floodplain inundation. This conclusion is too general as there is no any standard set for habitat suitability indices to quantify what percentage of flow variation affects the flora and fauna?

Response: We agree with the referee that this is a general conclusion. In fact, the scope and length of the manuscript doesn't allow detailed evaluation of the influence of the hydrological alteration onto the ecosystem service. This will be part of our future research in the pipeline, in that we will make use of the hydrological alteration to determine the influence on the hydrodynamic and morphology of the mayas, and then assess the impact on the ecosystem services. We have modified the conclusion to reflect limitation of this part of the analysis.

6. There is no any drawn conclusion from RVA analysis and also from change detected of precipitation and temperature. The rise in temperature may not or may be favorable for animals living in DNP......

Response: This is correct, there is no conclusion on the effect of hydrological alteration on the ecosystem services (vegetation, animals, etc.). This is beyond the scope of the paper. The paper focus on the quantification of the hydrological alteration only. Is it true or not, and how much?. These are essential information to evaluate the effect on the ecosystem services. The later needs a lot of work, first to define the ecosystem itself, and how it will be influenced, before quantifying the hydrological impact.

7. Figures are not well described or clarified in text. For example figure 4, 5 and 6 showed peaks or abrupt change points but nothing was mentioned what was happened during that time on the Mayas ecosystem or the historical impacts of the high and low flow. In general it lacks detail description of methodology and summarized result analysis and conclusion of the 33 parameters.

Response: The following paragraphs were added to the revised manuscript for more descriptions of the figures. "Figure 4 showed the Pettitt test results for the abrupt changing points. Significant abrupt changes for August flow (flood period) and for November flow (recession period) in Dinder River were observed. Decreasing of river flood directly affects inflows to mayas, which might lead to drying some of the mayas. It has been observed that during the early 1990s the area of some mayas inside DNP have radically decreased due to the variation in river discharge and sediment deposition processes (Abdel Hameed 1997). Such mayas can no longer store enough water to satisfy the needs of the wildlife populations throughout the dry season".

Figure 5 was described on page 10, line 26-30. Figure 6 was described on page 11, line 5-7. For more clarity, the description of Figure 6 was rewritten as "Figure 6 has shown that the post-impact median flow maxima for 1, 7, 30 and 90-day intervals in the Dinder river were, 14 %, 13%, 15%, and 14%, lower than pre-impact. In contrast, in the Rahad River increasing patterns were observed, with a post-impact median flow maxima for 1, 7, 30 and 90-day of 6 %, 9%, 16%, and 21%, respectively higher than pre-impact".

Detailed description of the IHA methodology was added to the manuscript (method section) as we pointed out in our response to comment 3. Summary of the result analysis and conclusion of the 33 parameters was improved by considering all comments and valuable suggestions by reviewers.

Some additional comments to specific paragraphs: a) page 4, line 25-30: The ESS should be clearly distinguished. Seperate them into the common categories, also by using a table.

Response: We have categories the ESS according to the categories proposed by the Mulinium Ecosystem Assessment (2005). The ESS were separated in table 1. Since this is a new table, we re-numbered tables in our revised manuscript.

b) page 5, line 21-23: Which loss of biodiversity occurred? Give facts!

Response: Here, we referred to literature on the effect of hydrological alteration on the ecosystem service , e.g., Sulieman and Mohammed (2014) who reported that aquatic and terrestrial plant species have disappeared and some are subjected to severe threats" as a result of destruction of their natural habitats. In our next research we plan to assess effect of hydrological alteration on ecosystem service.

c) page 7, line 30-32: Which kind of significance tests are used for the HA?

Response: To calculate the significance count for the deviation values, the IHA software developed by The Nature Conservancy (2009), is randomly shuffles all years of input

data and recalculates (fictitious) pre-impact and post-impact medians and coefficients of dispersions 1000 times. The significance count is the fraction of trials for which the deviation values of the medians or coefficients of dispersions were greater than of the real case. A low significance count (minimum value is 0) indicate that the difference between the pre and post-impact periods is highly significant, and a high significance count (maximum value is 1) indicate that there is little difference between the pre and post-impact periods. The significance count was interpreted similarly to a p value in MK statistics. This clarification was added to section 3.2 of the methods. A new table (Table 6) was also added to the manuscript to show the significant count for all parameters of interest.

d) figure 2: Which ET0 formula was used? How sensitive are the absolute ET0 values?

Response: The FAO-56 Penman-Monteith (PM) equation (Allen et al. 1998) was used to estimate ET0. Numerous studies have been performed using lysimeter data and have shown in most cases, the PM to be the best method for estimating ET0. A study by Droogers and Allen (2002) has assumed that the PM can be used to represent a standard for ET0 estimates, and proved that it is true in terms of practical applications found around the world. The PM was used with the worldwide data set with no adjustment.

e) figures 5 -7: The significance of the decrease / increase is not explained or tested. Moreover, there are some 0-values which are probably "no data" values (e.g. Rahad River in August).

Response: The first part is similar to comment c) which we have answered above. In second part of the comment, the referee is right, there is no records of flow at the beginning of the wet season for the months of July and August for Rahad River during 1999. Likewise, there are missing data in the records for the months of October and November (end of the wet season) for both Dinder and Rahad during 2002. Since there is only one hydrological station on the Dinder and only one station on the Rahad,

filling the missing data using regression is not possible. We can only fill in the missing data with the monthly mean values or leave it with no data. We have checked both options but we found no significant differences in the results since we are analysing more than 40 years of records. Thus, we have shown the results analysis of the data that contained no data for those particular months.

f) page 12, line 10-31: In this form not readable (you get "dizzy").

Response: Thanks. This section was summarized as "The RVA was originally designed for setting initial river management targets for river systems in which the natural hydrological regime has been altered by human activities. Significant alterations were reflected by the IHA parameters (median for a post-impact period) falling outside the range of variation observed for the period of record representing natural conditions. Thus, the intent of management targets derived using the RVA is for observed IHA parameter values to fall within a natural range of variation between high and low boundaries. Our results of the Dinder show that the magnitude of the monthly flows during the wet season (July-November) are within the range of variation except for November, the flow fall outside the high RVA boundary. In contrast, the Rahad river flows for all wet months fall outside the high RVA boundary (Fig 5). When analyzing the median flow maxima for Dinder, we found that flows for 1, 7, 30 and 90-day intervals, fall below the low RVA boundary. While in Rahad, all flow maxima fall within the RVA boundaries except for 30-day maximum the flow falls outside the high RVA boundary (Fig 6). The median flow rise rate for both Dinder and Rahad Rivers fall above the high RVA boundaries, while the median flow fall rate fall below the low RVA boundaries (Fig 7). All streamflow Hydrologic Alteration factors for the three categories of RVA for both Dinder and Rahad are reflected in Fig. 8.". Line 10-31 of the manuscript will be replaced by the above summary.

g) page 13, line 15-18: Quantify the LULC!

Response: Of course quantification of LULC will give much support to our result analysis, but requires a lot of work which is beyond the scope of this study and will be part of our next investigation.

Yours sincerely,

Khalid Hassaballah

On behalf of the co-authors

References:

Abdel Hameed SM, Awad NM, ElMoghraby AI, Hamid AA, Hamid SH, Osman OA. Watershed management in the Dinder National Park, Sudan. Agric For Meteorol. 1997;84(1):89–96.

Allen R.G., Pereira L.S., Raes D. & Smith M. 1998. Crop evapotranspiration: Guidelines for computing crop requirements. Irrigation and Drainage Paper No. 56, FAO, Rome, Italy.

Sulieman, H. & M. Mohammed, 2014. Patterns of woody plant species composition and diversity in Dinder National Park, Sudan. University of Kordofan Journal of Natural Resources and Environmental Studies 1(1):26-36.

The Nature Conservancy (2009). "Indicators of Hydrologic Alteration Version 7.1 User's Manual."

Droogers, P. and R. G. Allen (2002). "Estimating reference evapotranspiration under inaccurate data conditions." Irrigation and drainage systems 16(1): 33-45.

Please also note the supplement to this comment:
http://www.hydrol-earth-syst-sci-discuss.net/hess-2016-407/hess-2016-407-AC1-supplement.pdf

**Supplement:**

Table 1: The different ecosystem services provided by the DNP.

| Provisioning services | Regulating services | Supporting services | Cultural services |
|---|---|---|---|
| **Food:**
Mayas Ecosystem provides the conditions for growing food for both human and wild animals. Mayas provide fish for human consumption and grass for wild animals. Forests also provide food for human consumption such as honey. | **Local climate and air quality:**
Trees provide shade for wild animals whilst forests influence precipitation both locally and regionally. Trees or other plants also play an essential role in regulating air quality by removing pollutants from the atmosphere. | **Habitats for species:**
The DNP ecosystems provide different habitats for many individual plant or animal that are essential for a species' lifecycle to survive. Migratory species including mammals, birds and fish are all depend upon different ecosystems during their migrations. | **Tourism:**
High potential opportunity for tourism and education (e.g. attractive place for local people and foreigner and opportunity for research and training). Thus, it provides considerable economic benefits and is a potential source of income for the country. |
| **Fresh water:**
Mayas ecosystems play a vital role in the local hydrological cycle, as they regulate the flow and purification of water. Vegetation and forests influence the quantity of water available locally and further downstream. | **Carbon sequestration and storage:**
Ecosystems regulate the global climate by storing and sequestering greenhouse gases. In this way forest ecosystems in the DNP are carbon stores. Biodiversity also plays an important role by improving the capacity of ecosystems to adapt to the effects of climate change. | **Nutrient cycling:**
Mayas ecosystems regulate the flows and concentrations of nutrients through a number of complex processes that allow these elements to be extracted from their mineral sources or recycled from dead organisms. | **Aesthetic appreciation and inspiration for culture and art:**
The NDP Biodiversity, ecosystems and natural landscapes have been the source of inspiration for much of the art, folklore and culture in Sudan. |
| **Raw materials and Medicinal resources:**
The DNP ecosystems provide a great diversity of materials for construction and fuel including wood and charcoal. The DNP ecosystems also provide many plants used as traditional medicines for local people. | **Moderation of extreme events:**
The DNP plays an important roles in modulating the effects of extreme events. For example prevent or reduce flooding. Mayas wetlands attenuate floods by absorbing runoff peaks and storm surges. | **Maintenance of genetic diversity:**
Some habitats have an exceptionally high number of species which makes them more genetically diverse than others. | |

Table 3: Man-Kendall results of annual rainfall at the 12 examined precipitation stations.

| Station | Kendall's tau | S | P-value | Trend |
|---|---|---|---|---|
| Bahir Dar | -0.09898 | -107 | 0.3320 | No significant change |
| Gonder | -0.1176 | -156 | 0.02217 | No significant change |
| Samsam | -0.1979 | -111 | 0.1023 | No significant change |
| Umsienat | 0.1658 | 93 | 0.1728 | No significant change |
| Doka | 0.3333 | 187 | **0.0051** | Significantly increasing |
| Hawata | 0.2141 | 120 | 0.0777 | No significant change |
| Gedarif | -0.1260 | -755 | 0.0515 | No significant change |
| Gadambalyia | 0.0607 | 34 | 0.6247 | No significant change |
| Damazin | 0.3158 | 60 | 0.0537 | No significant change |
| Abu Naama | 0.2762 | 29 | 0.1659 | No significant change |
| Um Benien | 0.2952 | 31 | 0.1370 | No significant change |
| Sennar | -0.0533 | -228 | 0.4513 | No significant change |

Table 6: Medians and significant counts for IHA parameters for both Dinder and Rahad. Numbers in bold designate values that are statistically significant at 5% significant level.

| IHA PARAMETERS | Dinder | | | Rahad | | |
|---|---|---|---|---|---|---|
| | MEDIANS | | SIGNIFICANCE COUNT | MEDIANS | | SIGNIFICANCE COUNT |
| | Pre | Post | Medians | Pre | Post | Medians |
| **Parameter Group #1** | | | | | | |
| January | 0 | 0 | | 0 | 0 | |
| February | 0 | 0 | | 0 | 0 | |
| March | 0 | 0 | | 0 | 0 | |
| April | 0 | 0 | | 0 | 0 | |
| May | 0 | 0 | | 0 | 0 | |
| June | 0 | 0 | | 0 | 0 | |
| July | 43.4 | 49.82 | 0.6997 | 44.95 | 65.42 | **0.006006** |
| August | 266.3 | 210.4 | 0.5025 | 133.7 | 152.7 | **0.003003** |
| September | 292.1 | 297.6 | 0.8448 | 143.2 | 165.7 | **0.02503** |
| October | 101.2 | 89.91 | 0.4334 | 49.08 | 67.15 | 0.08308 |
| November | 0 | 14.19 | | 0 | 14 | |
| December | 0 | 0 | | 0 | 0 | |
| **Parameter Group #2** | | | | | | |
| 1-day maximum | 546.8 | 468.7 | **0.04304** | 166.6 | 175.8 | 0.2262 |
| 3-day maximum | 513.4 | 444.4 | 0.08308 | 164.4 | 175.5 | 0.1241 |
| 7-day maximum | 465.6 | 394.4 | 0.09209 | 159.1 | 173.1 | **0.04104** |
| 30-day maximum | 372 | 321.3 | 0.1832 | 143.9 | 166.9 | **0.01401** |
| 90-day maximum | 265.3 | 208.4 | 0.3213 | 113.7 | 137.7 | **0.01201** |
| Number of zero days | 250 | 233 | **0.01602** | 244 | 215.5 | **0.002002** |
| Base flow index | 0 | 0 | | 0 | 0 | |
| **Parameter Group #3** | | | | | | |
| Date of maximum | 244 | 250 | 0.3003 | 252 | 256 | 0.2893 |
| **Parameter Group #4** | | | | | | |
| High pulse count | 1.5 | 3 | 0.08509 | 2 | 2 | 0.4164 |
| High pulse duration | 63.5 | 11.5 | 0.1562 | 41 | 39.5 | 0.8679 |
| **Parameter Group #5** | | | | | | |
| Rise rate | 31.45 | 19.78 | **0.009009** | 4.56 | 2.67 | **0.04905** |
| Fall rate | -17.15 | -8.08 | **0.007007** | -4.37 | -2.435 | **0.01101** |
| Number of reversals | 39 | 43 | 0.06406 | 33 | 31 | 0.5495 |